# On the Bias of Next-Token Predictors Toward Systematically Inefficient Reasoning: A Shortest-Path Case Study

**Riccardo Alberghi**
riccardo.alberghi@studbocconi.it

**Elizaveta Demyanenko**
elizaveta.demyanenko@phd.unibocconi.it

**Luca Biggio**
luca.biggio@unibocconi.it

**Luca Saglietti**
luca.saglietti@unibocconi.it

## Abstract

Recent advances in natural language processing highlight two key factors for improving reasoning in large language models (LLMs): (i) allocating more test-time compute tends to help on harder problems but often introduces redundancy in the reasoning trace, and (ii) compute is most effective when reasoning is systematic and incremental, forming structured chains of thought (CoTs) akin to human problem-solving. To study these factors in isolation, we introduce a controlled setting based on shortest-path tasks in layered graphs. We train decoder-only transformers on question–trace–answer triples using a custom tokenizer, comparing models trained on optimal bottom-up dynamic programming traces with those trained on longer, valid traces involving backtracking. Surprisingly, with the same training-token budget, models trained on inefficient traces generalize better to unseen graphs. This benefit is not due to length alone—injecting arbitrary redundancy into reasoning traces fails to help and can even hurt performance. Instead, we find that generalization correlates with the model's confidence in next-token prediction, suggesting that long, coherent, and locally incremental traces make the training signal easier to optimize. The code is available at `https://github.com/riccardoalberghi/DP`

## 1 Introduction

Modern LLMs have made remarkable strides in tasks requiring reasoning and multi-step problem solving [1, 2, 3, 4]. Increasing evidence demonstrates that the performance of these models significantly improves when their reasoning unrolls in a step-by-step fashion, following CoTs reminiscent of how humans build their internal cognitive processes [5, 6, 7, 8]. Another milestone in the rapid development of LLMs is represented by the test-time-compute paradigm [9, 10, 11], driven by the intuition that harder problems often require more computational budget—and thus, longer CoTs. These insights have been central to the development of advanced reasoning agents capable of achieving unprecedented performance on complex tasks spanning mathematical problem solving [12, 13, 14], code generation [15, 16], and scientific inquiry [17, 18]. Despite such progress, reasoning remains an elusive concept—difficult to define precisely and challenging to study in the wild. How to teach machines to reason effectively, generalize across domains, and adapt to novel problems continues to be a fundamental open question.

Motivated by the need for a well-defined and interpretable setting to characterize reasoning in transformers, and inspired by [19], we turn to a controlled algorithmic task (see Fig 1): a synthetic shortest-path problem on layered graphs. Each problem instance—posed as a question to a transformer—consists of a source-to-target graph with integer edge costs; the model is tasked with

39th Conference on Neural Information Processing Systems (NeurIPS 2025).

answering with any minimum-cost path. While a bottom-up dynamic programming (DP) approach is the canonical optimal solution for this problem, several alternative strategies can also reach the correct answer. We design a family of such strategies and train transformers to follow them. This playground allows us to (i) generate possibly unlimited problem instances of varying levels of difficulty; (ii) probe key properties of effective reasoning agents identified in the literature. In particular, through this framework, we investigate the following questions:

**Q1.** Can a model learn to find an optimal path directly, without seeing intermediate steps?

**Q2.** Do intermediate algorithmic trajectories (CoTs) simplify the task for the model?

**Q3.** Is the optimal dynamic programming strategy the best CoT option to train the model?

**Q4.** When is increasing the number of CoT steps beneficial for the model?

**Q5.** Can different solution strategies have a more or less suitable structure for next-token prediction?

Our study suggests that, while globally optimal strategies may appear ideal for teaching transformers how to reason about the assigned problem, less efficient traces can be more in line with the inductive bias of next-token predictive architectures. Paradoxically, *inefficient reasoning* turns out to be *more effective*. In summary, we make the following contributions:

1. **Controlled reasoning benchmark**: We introduce a synthetic layered-graph shortest path task with a custom language format, enabling rigorous experiments on reasoning trace efficiency. The task serves as a proof-of-concept environment for studying how LLMs learn algorithms when provided different intermediate solution traces.

2. **Thinking step-by-step**: We find that training transformers to produce intermediate steps between question (graph instance) and answer (optimal path) substantially improves performance, in line with the behavior of modern LLM on problem-solving tasks.

3. **Efficiency vs. effectiveness analysis**: Through extensive experiments, we provide direct evidence that training on inefficient reasoning traces can improve model performance compared to training on optimal ones. This counterintuitive result holds even when trace lengths are equalized between conditions by adding redundant steps, highlighting that it is not merely the length of the reasoning trace, but its structure that matters.

4. **Next-token predictors favor inefficient traces**: We motivate our findings by measuring the confidence of trained models in predicting the next token. We find that this metric is higher for models trained on longer but systematic and locally incremental reasoning traces and lower on globally optimal strategies.

## 2 Problem Setup

As a testbed for our reasoning experiments, we consider a synthetic shortest-path problem in layered directed acyclic graphs (DAGs) with integer edge costs. We generate random graph instances from a family with parameters $\{L, K, C, p_e\}$, respectively representing: the maximum number of layers, the maximum number of nodes per internal layer, the maximum edge cost, and the average connectivity between nodes in successive layers. For simplicity, the first and last layers contain exactly one node, the source and the destination of the sought path. We also ensure that no nodes are either completely disconnected from the previous layer or have no connections to the next one. Once the graph has been defined, we label the nodes in a top-to-bottom and left-to-right order. See Fig. 1 for an example of a small problem instance.

**Dynamic programming solution.** The above-described task can be framed as a simple multi-step reasoning task, involving the successive solution of a set of sub-problems, i.e., the shortest path from the source node to all intermediate nodes. The goal of the reasoner is to adopt an efficient strategy for completing all the required reasoning steps and building an optimal solution. Enumeration of all partial paths would entail a $\mathcal{O}\left(K^L\right)$ computational cost, yet the cost can be cut down to $\mathcal{O}(LK^2)$ if the sub-problems are conveniently ordered and the partial solutions are stored away. A bottom-up dynamic programming (DP) approach, which solves the sub-problems in layer order—from shorter to longer partial paths, yields an optimal shortest path with the minimum number of reasoning steps.

### 2.1 Tokenization

We aim to solve the shortest-path problem with a GPT-like model, trained from scratch on next-token prediction over a set of question-trace-answer examples. We define a task-specific token dictionary,

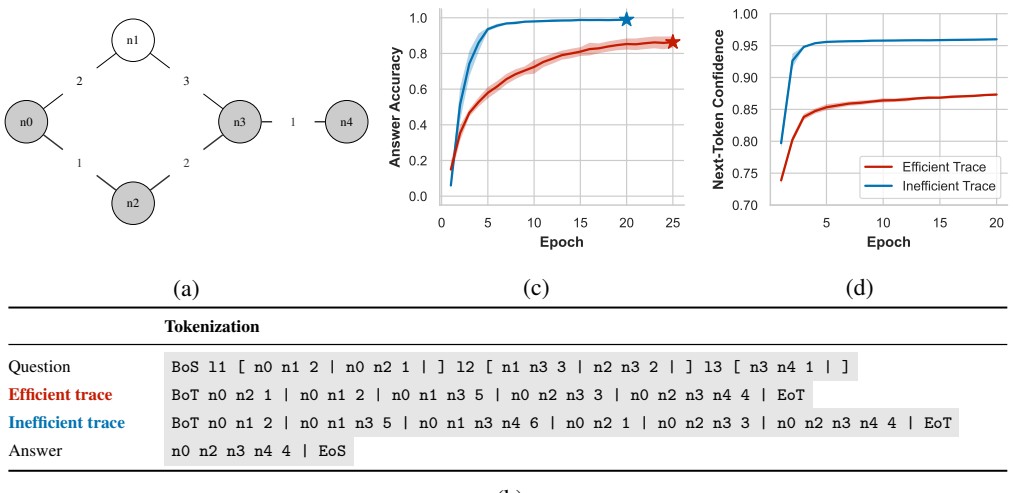

|  | Tokenization |
|---|---|
| Question | `BoS 11 [ n0 n1 2 | n0 n2 1 | ] 12 [ n1 n3 3 | n2 n3 2 | ] 13 [ n3 n4 1 | ]` |
| **Efficient trace** | `BoT n0 n2 1 | n0 n1 2 | n0 n1 n3 5 | n0 n2 n3 3 | n0 n2 n3 n4 4 | EoT` |
| **Inefficient trace** | `BoT n0 n1 2 | n0 n1 n3 5 | n0 n1 n3 n4 6 | n0 n2 1 | n0 n2 n3 3 | n0 n2 n3 n4 4 | EoT` |
| Answer | `n0 n2 n3 n4 4 | EoS` |

(b)

Figure 1: **Reasoning in the shortest-path problem.** (a) Example of a layered DAG with integer edge costs. The nodes are labelled top-to-bottom, left-to-right, 'n0' and 'n4' representing the source and the destination of the path. The nodes in the optimal path, with a cumulative cost $1 + 2 + 1 = 4$, are highlighted in grey. A human reading the graph would backtrack before discovering the cheaper 'n0→n2→n3→n4' path—mirroring the strategy the model ultimately favors. (b) We introduce custom tokens to uniquely represent the graph structure (question), the trace of the solver algorithm (efficient/inefficient trace examples), and the corresponding shortest-path solution (answer). (c) Generalization performance, measured in terms of the probability of returning an optimal path in the answer, of a model trained on optimally efficient (dynamic-programming like) traces and a model trained on inefficient (depth-first-search like) traces, with a corpus of 200K question-trace-answer examples. The inefficient reasoning traces are roughly 75% longer than the efficient ones. Only the model trained on the inefficient traces robustly generalizes to unseen graphs. (d) Next-token confidence of models trained on efficient vs inefficient traces, displaying an inductive bias of the transformer toward learning the latter.

with unique tokens representing: begin-sequence ( `BoS` ) and end-of-sequence ( `EoS` ); begin-of-think ( `BoT` ) and end-of-think ( `EoT` ); each possible layer label; each possible node label; each integer in the range of possible cumulative edge costs; a separator token, `|` ; two additional syntax tokens, `[` and `]` . Each part of the question-trace-answer triples follows well-defined syntactic rules:

**Question**: Opening with a `BoS` token, the question contains a complete token encoding of the graph. As in the example in Fig. 1(b), each layer of the graph is represented by a sequence of tokens, opening with the layer label, followed by the corresponding edge list enclosed between `[` `]` tokens. Each edge is declared as a pair of node labels, the corresponding cost, and a separator.

**Trace**: The reasoning trace, enclosed between `BoT` - `EoT` tokens, contains a set of reasoning steps, each represented as a succession of node labels, the cumulative cost of the associated path, and a separator `|` , as shown in 1(b). In Sec. 3, we will describe in detail the family of traces considered in this work and how we parametrically control their length and efficiency. Crucially, *all training traces meet the following optimality criteria*: i) The reasoning steps always contain correct path-cost statements; ii) Longer paths are built incrementally, adding a single node to a previously seen partial path; iii) Once a better alternative is found, sub-optimal partial paths are never considered as building blocks for longer paths; iv) All reasoning traces are complete and deterministically allow the construction of an optimal path. Note that, by writing down the intermediate shortest-path solutions, the reasoning trace can be seen as an explicit tabulation of the optimal partial costs of the shortest-path problem, and can be leveraged to avoid the exponential computational cost in the number of graph layers.

**Answer**: The answer is simply a repetition of the optimal path found during the reasoning trace, and follows the same syntax: a succession of nodes – from source to destination nodes, the associated

cost, and a separator `|` . The generated sequence is then closed by a `EoS` token. An example can be seen in Fig. 1(b).

## 3    Experimental Setup

We leverage the controlled nature of our problem setting to define a random generator of reasoning traces with a tunable degree of efficiency. These exact traces are then used to build a training corpus of question-trace-answer triples, and train a next-token prediction model on the shortest-path task.

The efficiency level of the CoTs is uniquely determined by the exploration order according to which the model computes partial paths and costs and then composes them to build the optimal path. As the trace generator traverses the graph, it maintains an exploration queue containing path continuations yet to be considered. The associated priority weights depend on a temperature parameter, the *efficiency* $\eta$, controlling whether shorter vs. longer partial paths should be explored first. When multiple partial paths of the same length can be continued, their relative order is fully randomized. The internal logic of the exploration algorithm and its dependence on $\eta$ can be seen in Fig. 2(a).

Thus, $\eta$ can bias the underlying algorithm toward a layer-by-layer (positive $\eta$) or a depth-first (negative $\eta$) approach. This directly affects the number of reasoning steps, since previously explored paths will need revising if a better route to an intermediate node is encountered. For this reason in the paper we refer to positive $\eta$ values as efficient (less total steps) and to negative $\eta$ values as inefficient (more total steps). Note, however, that the *trace optimality criteria* described in section 2.1 guarantee that the number of steps remains polynomial (worst-case $\mathcal{O}(CL^2K^2)$[1]), as evidenced from the table in Fig. 2(b).

Furthermore, we can inject reasoning *redundancy*, by artificially increasing the length of the trace via repetition of full reasoning steps. To study the importance of preserving the CoT structure, we consider two variants: a *deterministic* version, where each reasoning step is immediately repeated and then never revisited and a *randomized* version, where completed reasoning steps are re-appended to the exploration queue with probability 1/2.

**Types of reasoning traces.**    To simplify the interpretation of the results, we will consider a few prototypical settings for the reasoning trace generator:

- $\eta = +5$ (DP): at this temperature, the reasoning trace corresponds to a bottom-up DP trace, systematically exploring the graph in a layer-by-layer order.

- $\eta = 0$: the reasoning trace chooses the next path to be explored uniformly at random among the available options, irrespective of the path length, and might include some backtracking.

- $\eta = -5$ (DFS): the reasoning trace systematically prioritizes a depth-first approach, requiring substantial backtracking.

- $\eta = +5$ (DR): each reasoning steps is deterministically repeated twice in a row.

- $\eta = +5$ (RR): each reasoning step, after completion, is re-added to the exploration queue with probability $1/2$. Note that this implies that, in expectation, each reasoning path is repeated twice.

Note that, given the exponential law employed to sample the layer order, the efficiency values $\eta = \pm 5$ are large enough to fully order the exploration protocol (effectively behaving as $\eta = \pm\infty$, but avoiding numerical instabilities). As shown in Fig. 2(b), the average length of the traces increases with lower values of the efficiency $\eta$, changing by roughly 75% when switching from $\eta = +5$ (DP) to $\eta = -5$ (DFS). In the redundancy cases, the number of repetitions is matched between the randomized and deterministic versions. Examples of the different trace types are provided in the Supplementary Materials (SM).

**Trained model.**    We train from scratch a Phi3 [20] small language model, with 3 layers, 12 attention heads, 768 hidden dimensions, and 28.5M total parameters, for the next-token prediction task on the procedurally generated training examples. During training, we mask out the question (i.e., the

---

[1]Given the structure of the exploration algorithm Fig. 2(a), each one of the $\mathcal{O}(LK)$ nodes can be at most revisited $\mathcal{O}(CL)$ times, since a best cost improvement is needed to trigger backtracking.

```
 1: Initialize empty queue and weight lists, $\mathcal{E}$ and $\mathcal{W}$
 2: Initialize best cost and best path dictionaries, $\mathcal{C}$ and $\mathcal{P}$
 3: for $n \in$ layer1 do
 4:     append($\mathcal{E}$, $(n0, n, 0)$); append($\mathcal{W}$, 1)
 5: end for
 6: while $\mathcal{E} \neq \emptyset$ do
 7:     choose( (src,dst,layer) from $\mathcal{E}$, w.p $\propto \mathcal{W}$)
 8:     c, path $\leftarrow \mathcal{C}$(src) + cost(src,dst), $\mathcal{P}$(src) $\cup$ dst
 9:     if $c < \mathcal{C}$(dst) then
10:         $\mathcal{C}$(dst), $\mathcal{P}$(dst) $\leftarrow c$, path
11:         for $n \in$ destinations(dst) do
12:             e, w $\leftarrow$ (dst,n,layer+1), $\exp(-\boldsymbol{\eta}(\text{layer}+1))$
13:             if e $\notin \mathcal{E}$ then
14:                 append($\mathcal{E}$, e); append($\mathcal{W}$, w)
15:             end if
16:         end for
17:     end if
18: end while
```

| type | CoT steps |
|---|---|
| $\eta = +5$ | $33 \pm 20$ |
| $\eta = 0$ | $43 \pm 33$ |
| $\eta = -5$ | $58 \pm 54$ |
| $\eta = +5$ (RR) | $65 \pm 41$ |
| $\eta = +5$ (DR) | $65 \pm 40$ |

(a)                                                   (b)

Figure 2: **Impact of the efficiency $\eta$.** (a) The exploration algorithm used for determining the shortest-path. The exploration order depends on the parameter $\eta$ (line 12). (b) The effect of $\eta$ on the distribution of number of reasoning steps (estimated on 100K independent samples).

graph information) and the `PAD` tokens from the loss function, so that the model only learns to predict traces and answers within the context of the question. We employ a constant learning rate of $2 \times 10^{-5}$, and a batch size of $\sim 16$K tokens (excluding the `PAD` tokens for a fair comparison between efficiency levels). In SM, we show how adding layers to the transformer architecture can impact performance and sample efficiency.

Given the low entropy of our custom language and the mathematical nature of the reasoning task, we default to zero-temperature generation, greedily choosing the maximum likelihood next token, unless otherwise stated.

**Metrics and performance evaluation.** We implement a parser able to evaluate the sequence of tokens produced by the model and return a set of metrics on the quality of the generated trace and answers. The accuracy of the answers and efficiency of the trained models are assessed via two main indicators:

- **answer accuracy**, constructed by requiring the optimality of the path in the answer, i.e., the conjunction of: i) the path is possible, involving connected nodes; ii) the path has the expected length, i.e. the number of layers in the graph; iii) the declared cost is minimal, as obtained with a DP solver; iv) the path and cumulative cost declarations are consistent.
- **number of reasoning steps**, counting the number of well-formatted partial-path statements.

We also check for multiple secondary metrics on the quality of the reasoning steps in the trace, including: i) if they contain syntax errors; ii) if they are incremental; iii) if they only build upon optimal partial paths; iv) if the path-cost declaration is consistent; v) if the reasoning steps are repeated. To simplify the exposition, the analysis of these metrics is deferred to SM.

Finally, inspired by [21], we also measure the **next-token confidence** of the model along the reasoning trace and answer, i.e., the average sampled token probability. Note that here we find this metric to be equivalent to the min-margin metric suggested in [21], in agreement with a footnote observation therein.

If not otherwise specified, the metrics are averaged over a test set containing new graphs with sizes in the training graph distribution, and reported with 1-sigma error bars across five random training seeds. The best generalization accuracies are denoted with a star symbol in the figures.

## 4 Results

We aim to characterise the impact of the presence of the reasoning trace, and of its length and structure, on the performance of our model on unseen shortest-path problems. In the following, we fix the parameter settings for the graph generator to $\{L = 7, K = 6, C = 5, p_e = 0.6\}$, and ensure that all graph examples in training and test sets are unique.

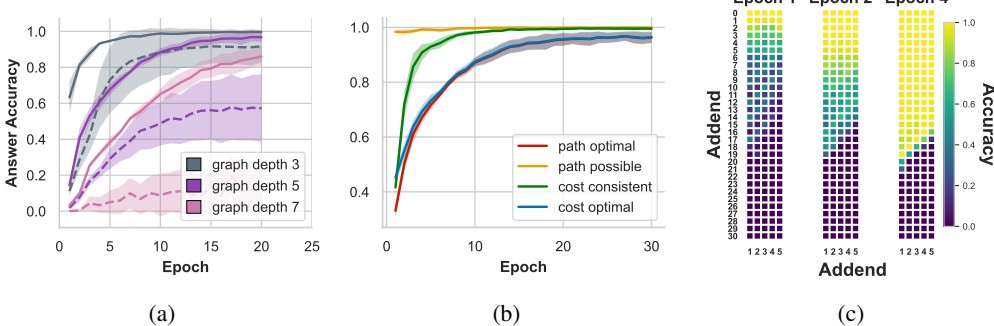

Figure 3: **Learning to find the shortest path.** (a) Generalization performance of two models trained on $\sim 340K$ graphs, respectively without reasoning traces (*dashed*) and with the $\eta = +5$ (DP) traces (*full*), over graphs with depths 3-5-7. (b) Progress on intermediate training goals for the $\eta = +5$ (DP) model. (c) Acquisition of the integer addition sub-task, during the $\eta = +5$ (DP) training. The plot shows the probability of the model predicting the correct $row + column$ sums at different epochs.

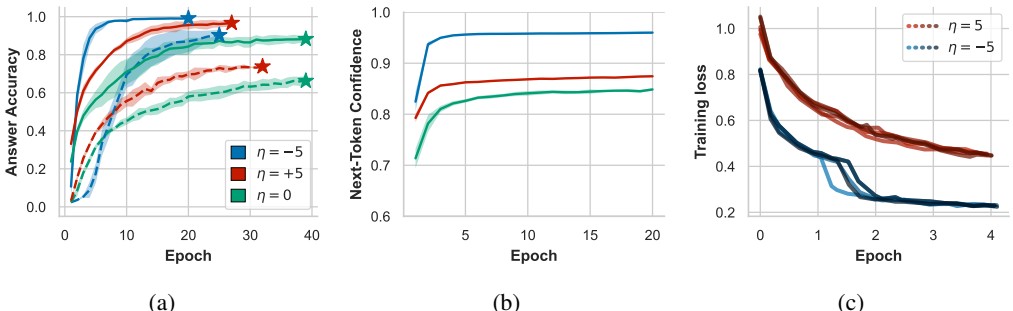

Figure 4: **Impact of trace efficiency.** (a) Comparison of the generalization performance between models trained on efficient $\eta = +5$ (DP), intermediate $\eta = 0$, and inefficient $\eta = -5$ (DFS) traces, with a training token budget of 32M (*dashed*) and 128M (*full*) tokens. (b) Next-token confidence measured on the test set of models trained on efficient $\eta = +5$ (DP), intermediate $\eta = 0$, and inefficient $\eta = -5$ (DFS) traces with a training token budget of 128M. (c) Training losses for 5 different seeds of $\eta = -5$ (DFS), showing sudden jumps at the beginning of the 2nd epoch, and of $\eta = +5$ (DP), where optimization is slower and more continuous.

**Finding the shortest path with a next-token predictor (Q1-2).** The first finding we report on, shown in Fig. 3(a), is that a next-token predictor can learn to solve shortest-path problems in moderate-sized graph instances, when sufficient training data is presented (here $\sim 340K$ graph examples). However, good generalization on the larger in-distribution graph instances can only be achieved when the model is allowed to produce a reasoning trace before returning an optimal path: the solid curves show the performance, on $3, 5, 7$ layer graphs, of a model trained in the dynamic programming limit $\eta = +5$ (DP). Instead, the dashed curves show that a model trained on the question-answer task, with the same number of examples, is unable to learn a generalizable rule for solving previously unseen large problems.

In Fig. 3(b), we break down the learning process for the $\eta = +5$ (DP) model, showing the pace at which intermediate learning goals are unlocked, such as proposing candidate solutions that are possible, or correctly associating paths and costs. In Fig. 3(c), we study how the model acquires the sub-task of integer addition (here restricted to a small subset $\mathcal{O}(CL)$ of possible cumulative sums). In the figure, we measure the probability of the model assigning the maximum logit to the correct sum token, and display the order in which the different additions are learned during training. Note that this subtask is not trivial [22, 23], although the training set extensively covers the relevant range of cumulative costs (see SM), since the model has to understand where to pick up the partial costs to be added from the question and the previous steps in the trace.

**Impact of efficiency and structure of the reasoning trace (Q3-5).** We explore the impact of the reasoning trace efficiency on model performance, at fixed token budget by comparing 32M tokens with 128M tokens. In Fig. 4(a), the best accuracy in predicting the shortest path is achieved by the $\eta = -5$ (DFS) model, with the longest reasoning traces and systematic backtracking, followed by the DP-like $\eta = +5$ (DP), and finally by the $\eta = 0$ models. Note that, since the token budget at training is fixed, the $\eta = -5$ (DFS) inefficient model sees about $1/3$ fewer graph examples compared to the efficient one at $\eta = +5$ (DP), yet absorbs the training set information more effectively. Moreover, as shown in Fig. 2(b), the traces for $\eta = 0$ are longer than those of $\eta = +5$, yet the corresponding curves are sub-optimal—highlighting that higher test-time compute, represented by longer traces, does not necessarily translate into better performance. In Fig. 1(c), we also train the $\eta = \pm 5$ models with an equal number of graph examples ($\sim 200K$, as in the 128M dataset for $\eta = -5$), finding an even larger performance gap.

In Fig. 4(b), we find a plausible explanation of the effectiveness of $\eta = -5$ and the poor performance of $\eta = 0$, looking at the next-token confidence [21] of the three models. While the $\eta = 0$ traces are longer than the $\eta = +5$ (DP) ones, the associated flat distribution over the exploration order, mixing depth-first and layer-by-layer exploration, reduces the confidence of the model in predicting the next step. This, in turn, undermines the learning effectiveness of the model trained on this trace type.

We can precisely quantify the degeneracy of the exploration order for each value of $\eta$, by computing the average surprisal $\mathcal{S}$ (i.e., the Shannon information) associated with the selection of the next path from the exploration queue. We obtain

$$\mathcal{S}_{\eta=+5} = 1.3262 \pm 0.0006, \qquad \mathcal{S}_{\eta=-5} = 0.4821 \pm 0.0002, \qquad \mathcal{S}_{\eta=0} = 1.905 \pm 0.003,$$

confirming that for $\eta = 0$ the order of the reasoning steps is the most uncertain. On the other hand, both $\eta = +5$ (DP) and $\eta = -5$ (DFS) strategies share a deterministic approach in the layer exploration order, but the degeneracy of equivalent choices is higher for $\eta = +5$ (DP). A similar study of the gap in next-token confidence, but for a fixed number of training graph examples, is shown in Fig. 1(d).

The effect of trace predictability can also be seen from the optimization trajectories, in Fig. 4(c). The $\eta = -5$ (DFS) trajectories often exhibit a sudden jump in the training loss, occurring around the beginning of epoch 2, which highlights a "eureka" transition in the interpretation of the reasoning trace examples. We hypothesize this sudden transition could be explained by the emergence of specific circuits within the model forward pass, similar to those identified by [24, 25, 26, 27, 28]. We therefore see the mechanistic interpretability analysis of our trained model as an interesting avenue for future work.

**Injecting redundancy into the reasoning traces (Q4).** To further explore the impact of increased test-time compute, in the absence of stochastic confounders that decrease the predictability of the trace, we compare the $\eta = +5$ (DP) baseline with a model trained on deterministically redundant $\eta = +5$ (DR) traces. Since we repeat each reasoning step twice in a row, in principle, the model can build a mechanism for choosing the next path that relies on this repetition for artificially increasing the test-time compute. Note that the elongated CoTs have more steps than $\eta = -5$ (DFS) on average. In Fig. 5(a), we show that artificially increasing the length of the reasoning traces, without a systematic change in the exploration strategy, induces a slight performance deterioration if trained with a fixed token budget (128M). This aligns with recent research showing that CoT length is task-dependent [29] and that the global *structure* of the CoT is often more important than its content [30].

**Bias towards longer CoTs (Q5).** In the previous experiments, a non-systematic structure in the reasoning trace—as in the case of $\eta = 0$—was associated with a reduced capability of the model to predict the next token, affecting both optimization and generalization performance. Structural perturbations to the reasoning traces can, in fact, strongly hinder performance [31]. A similar effect can be seen if the redundancy is introduced in a randomized fashion. In Fig. 5(a), we show the probability of producing a correct shortest path solution for a model trained in the $\eta = +5$ (RR) setting, at zero sampling temperature. Apart from the reduction in the model accuracy compared to the deterministic analogue, in Fig. 5(b) we also observe that the generated trace length initially diverges from the expected one: the model enters repetition loops that can elongate the CoT indefinitely. Longer training is required for regularizing this behavior and eventually attaining better accuracy.

A similar high-verbosity tendency can be traced in the $\eta = 0$ case, in Fig. 6(a). At zero sampling temperature, the model initially gravitates towards the trace lengths of $\eta = -5$ (DFS), gradually

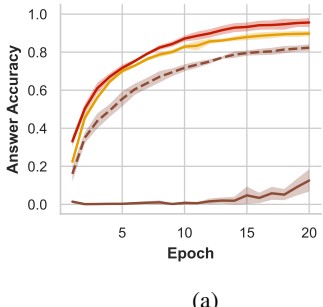

| temperature | path optimality (%) | CoT steps |
|---|---|---|
| 0.0 | 8 | $244 \pm 182$ |
| 0.2 | 37 | $177 \pm 132$ |
| 0.5 | 81 | $89 \pm 63$ |
| 0.7 | 85 | $77 \pm 51$ |
| 1.0 | 84 | $72 \pm 44$ |
| 1.2 | 79 | $71 \pm 43$ |
| 1.5 | 62 | $68 \pm 42$ |
| 1.7 | 44 | $67 \pm 43$ |
| 2.0 | 20 | $62 \pm 43$ |

(a)                                      (b)

Figure 5: **Redundant traces.** (a) Comparison of generalization performance between models trained on traces with efficiency $\eta = +5$ (DP), $\eta = +5$ (DR), and $\eta = +5$ (RR) (with sampling temperatures $T = 1$ (*dashed*) and $T = 0$ (*full*)), trained with a 128M token budget. (b) Regularization effect of sampling temperature on $\eta = +5$ (RR), where the answer accuracy improves and the average CoT length converges to the expected one from training data at higher temperatures.

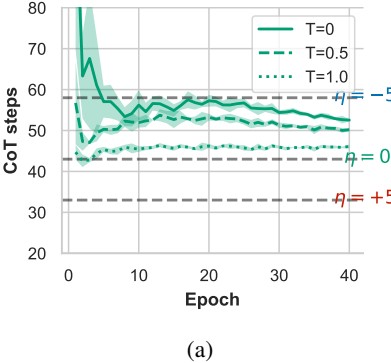

| temperature | path opt. (%) | CoT steps |
|---|---|---|
| *Epoch 20* | | |
| 0.0 | 79 | $59 \pm 49$ |
| 0.5 | 86 | $52 \pm 41$ |
| 1.0 | 84 | $45 \pm 32$ |
| 1.5 | 64 | $44 \pm 31$ |
| *Epoch 40* | | |
| 0.0 | 84 | $52 \pm 41$ |
| 0.5 | 87 | $50 \pm 38$ |
| 1.0 | 85 | $46 \pm 34$ |
| 1.5 | 74 | $44 \pm 31$ |

(a)                                      (b)

Figure 6: **Impact of sampling temperature.** (a) The length of the CoTs produced by $\eta = 0$ model (*full*) initially converges to the expected length of the inefficient traces, $\eta = -5$ (DFS), gradually recovering after many epochs. By sampling at positive temperature (*dashed and dotted*), the length converges to the expected one for $\eta = 0$. (b) While converging to the expected number of reasoning steps for the $\eta = 0$ strategy, the $\eta = 0$ model also achieves better answer accuracy at non-zero temperatures.

recovering after many epochs. In both cases, the models display a bias towards mechanisms that systematically induce longer traces. This finding is consistent with multiple works showing the bias of LLMs for high verbosity [2, 32, 33].

**Sampling at non-zero temperature.** To facilitate the imitation of stochastic behaviors for the $\eta = 0$ and $\eta = +5$ (RR) models, we try exploring the effect of raising the sampling temperature. Counter-intuitively, larger temperatures are found to regularize the verbosity of the generated sequences instead of encouraging it, as already noted in [34]. In Fig. 5(b) and Fig. 6(b), the best performance of the two models is recorded when the reasoning trace lengths become more compatible with those of the corresponding training examples.

## 5    Related Work

Transformer-based models such as OpenAI-o1 [1] and DeepSeek-R1 [2] have achieved state-of-the-art results on tasks involving mathematical reasoning and logical inference [10, 35]. Much of their success is attributed to the use of CoT reasoning and compute scaling at inference time. The role and limitations of these components have become active areas of empirical and theoretical investigation. Several studies have shown that including intermediate CoT steps significantly enhances transformer performance [36, 8, 37, 38]. For instance, [8, 38] demonstrate that the expressive power of decoder-only transformers increases with the length of CoT sequences. Similarly, [37] finds that, in parity

problems, CoTs not only boost expressiveness but also improve sample efficiency. Our experiments corroborate these findings, showing that training with CoT generally improves performance. However, we also observe that not all CoTs are equally beneficial, and longer traces do not always yield better outcomes.

This work aims to explore reasoning in a controlled yet nontrivial setting. Graph-based algorithmic tasks serve as an effective benchmark for evaluating whether models can learn structured reasoning and generalize beyond the training distribution [39, 40, 41, 42, 43]. For example, [39] train Graph Neural Networks to replicate intermediate steps of classical algorithms. Meanwhile, [42] investigate transformer performance with respect to architecture parameters and the use of scratchpad tokens [36]. Most relevant to our setting, [43] demonstrate that autoregressive transformers often struggle to generalize on the seemingly simple path-star task, frequently relying on heuristics such as the Clever Hans effect. In contrast, we find that introducing intermediate reasoning steps significantly enhances next-token prediction accuracy and confidence.

Our problem setup is motivated by recent work examining transformers' reasoning and planning capabilities through prediction of $A^*$ search dynamics [19, 44]. Like these studies, we task models with generating both a reasoning trace and a final answer, given a structured graph as input. While prior work has shown the benefits of CoT training in terms of sample efficiency, we further demonstrate that the structure of the CoT plays a critical role in performance. We introduce a confidence-based heuristic to evaluate the robustness of generated traces. To develop this heuristic, we build on the findings of [21], who propose test-time decoding metrics based on top-token probability or the gap between the top two probabilities. We apply the former to show that some reasoning traces yield more confident next-token predictions. Our results align with [45], who use confidence scores to guide the compression of redundant CoTs. Finally, recent studies show that transformers can learn to generate long CoTs via supervised learning [10, 30]. Notably, [30] emphasize that CoT structure can be more important than content—a finding that supports our observation that models trained on structured but suboptimal algorithmic traces outperform those trained on optimal yet less interpretable ones.

# 6  Discussion

Our controlled study reveals three high-level takeaways: (i) *Chain-of-thought is pivotal.* When the model is forced to jump directly from question to answer, performance on larger graphs collapses, while a well-structured trace restores strong generalisation. (ii) *Structure beats global optimality.* Training on longer depth-first traces that revisit nodes consistently outperforms training on globally optimal dynamic-programming traces, despite seeing fewer unique graphs under the same token budget. (iii) *Next-token confidence is a good proxy for learnability.* Across all settings, higher average top-token probability along the trace correlates with answer accuracy, suggesting that "easier-to-predict" reasoning signals drive sample-efficient learning. Our findings caution that what seems most sensible to *teach*—the shortest, globally optimal trace—is not what next-token predictors *learn* most readily; they favour systematic, locally incremental yet longer reasoning paths.

**Limitations.**   The presented setting, deliberately minimalist, entails several caveats. Our experimental design is built around a synthetic algorithmic task expressed in a custom token language, and the extent to which the observed biases transfer to natural-language or multimodal problem settings remains to be investigated. Different algorithmic domains (sorting, SAT, theorem proving) could yield different optimal–inefficient trade-offs, and different model architectures might display inductive biases that lead to different conclusions compared to auto-regressive models. We believe the exploration of these themes is an exciting direction for future work.

**Future work.**   A natural next step is to explore whether transformers can be steered toward more compact reasoning: curriculum schedules that progressively shorten traces, or reinforcement-learning objectives that penalize verbosity, might encourage the model to internalize a leaner algorithm without sacrificing accuracy. The benchmark's tunable difficulty and transparent structure also lend themselves to a mechanistic interpretability analysis; the attention patterns and hidden activations could reveal computational circuits that implement incremental cost aggregation versus backtracking. Finally, the efficiency-versus-effectiveness trade-off merits further investigations in larger language models and natural-language tasks.

## 7 Broader Impact

Our work explores how the inductive biases of decoder-only transformer models influence their reasoning abilities in a fully controlled setting. As reasoning in large language models remains a critical and evolving area of research in modern AI, we aim for our findings to inspire further investigation and support efforts to better understand and steer the behavior of contemporary AI systems.

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

# A Experimental details

## A.1 Training and testing configuration

We run our experiments on a Phi3 [46] architecture, in a consistent hyperparameter setting specified in Section 3. During training, we employ the AdamW optimizer, with weight decay of $0.1$, constant learning rate $2 \times 10^{-5}$, and momentum parameters $\beta_1 = 0.9$ and $\beta_2 = 0.999$. Training runs are terminated when the test loss exhibits a consistent increase after plateauing in the minimum.

Batch sizes are defined in terms of total token count (excluding `PAD` tokens to ensure fair comparison across different efficiency levels $\eta$), rather than by the number of samples, and are fixed at $16384$ tokens per batch. The context length is set to $4096$ tokens for experiments involving reasoning traces, and to $256$ tokens for standard question-answer tasks.

The codebase uses the standard PyTorch and HuggingFace libraries [47, 48], with all unspecified parameters set to their default values. Model performance is evaluated post-training using checkpoints saved at the end of each epoch. All predictions are generated using the vLLM inference framework [49].

With precision being limited to FP16, in a configuration with 3 layers the memory footprint typically reaches approximately 14GB of GPU memory. Most training runs have been executed on hardware configurations featuring either an Nvidia A100 GPU with 80GB memory or an RTX4090 GPU with 24GB memory, each accompanied by 32 CPU cores.

## A.2 Data generation

As described in in Section 2, each datapoint represents a layered directed acyclic graph (DAG). Each graph instance is randomly sampled from a distribution parameterized by the tuple $\{L, K, C, p_e\}$, where $L$ is the maximum number of layers (excluding the source), $K$ is the maximum number of nodes in any internal layer, $C$ is the maximum possible edge cost (so that costs are integer-valued and lie in $\{1, \ldots, C\}$), and $p_e$ controls the expected edge density between successive layers.

To generate a graph, we first sample the number of layers $\tilde{L}$ uniformly from the set $\{2, \ldots, L\}$. The first and last layers contain exactly one node each, representing the source and the destination of the graph. For each internal layer $\ell = 1, \ldots, \tilde{L} - 1$, we sample its number of nodes uniformly from $\{2, \ldots, K\}$.

Let $\mathcal{V}_\ell$ be the set of nodes in layer $\ell$. For each consecutive pair $(\mathcal{V}_\ell, \mathcal{V}_{\ell+1})$ we construct a weighted adjacency matrix

$$A^\ell \in \mathbb{R}^{|\mathcal{V}_\ell| \times |\mathcal{V}_{\ell+1}|}.$$

With probability $p_e$, an entry $A^\ell_{ij}$ is assigned a cost drawn uniformly from $\{1, \ldots, C\}$; otherwise $A^\ell_{ij} = +\infty$.

After generating $\{A^\ell\}_{\ell=1}^{\tilde{L}-1}$, in order to guarantee the existence of at least one valid path from the source to the destination, we enforce two simple connectivity constraints:

1. Any node that is not in the final layer and has no outgoing edges is connected to a random node in the next layer, with an edge cost drawn uniformly in $\{1, \ldots, C\}$.

2. Any node other than the source that has no incoming edges is connected to a random node in the previous layer, again with a uniformly sampled cost.

Once the graph construction is finalized, it is serialized into a deterministic token sequence using a custom tokenizer. In this way we generate a set of unique sequences, ensuring that no graph instance is repeated while allowing for semantic graph similarities (e.g. instances that are identical up to a permutation of nodes). Finally, the resulting corpus of tokenized sequences is split into training and test sets using a 9:1 ratio.

## A.3 Evaluation metrics

To track progress on intermediate training objectives, we compute a range of CoT- and answer-level metrics. While some of these were already introduced in Section 3, Table 1 provides a complete description of all metrics used for performance evaluation.

| Metric | Type | Category | Description |
|---|---|---|---|
| is possible | bool | Answer | Whether the path in the answer contains valid nodes for each layer and follows the correct layer order. |
| is cost consistent | bool | Answer | Whether the cumulative cost equals the sum of edge weights along the path. |
| is cost optimal | bool | Answer | Whether the reported cost equals the globally optimal one. |
| length is correct | bool | Answer | Whether the path has exactly one node per layer. |
| is correct (or **answer accuracy**) | bool | Answer | Whether the answer satisfies all A-level criteria. |
| Number of steps | int | CoT | Number of reasoning CoT steps (delimited by \|). |
| Repeated steps | int | CoT | Count of CoT steps that repeat a previously seen sub-path. |
| Possible sub-paths | int | CoT | Count of CoT steps that represent valid sub-paths. |
| Consistent steps | int | CoT | Count of CoT steps whose cost matches the sum of the corresponding sub-path costs. |
| Subproblem optimal steps | int | CoT | Count of CoT steps that consider only current best sub-paths. |
| Steps with a skipped subproblem | int | CoT | Count of CoT steps that contain nodes to which current best cost is not known. |
| Syntax errors | int | Both CoT & Answer | Number of structural or token-level errors. |

Table 1: Evaluation metrics with their associated category (Answer, CoT, or both).

# B  Additional results

## B.1  Quality of model-generated reasoning traces

In the main text, we focused on the capability of our trained models to provide a correct answer to the proposed shortest-path questions, and showed that with enough data they approach perfect accuracy. However, we have not analyzed the quality of the produced reasoning traces while this accuracy is attained. In Fig. 7, we compare several CoT-level metrics between models trained on traces with efficiency $\eta = -5$(DFS) and $\eta = +5$(DP). Overall, we find that the models are able to perfectly absorb the syntax and algorithmic logic of the training examples, mostly producing perfect traces (in correspondence to the correct answers) with the expected systematic exploration order.

However, Fig. 7(a) reveals a possible origin of the performance gap observed between $\eta = +5$(DP) and $\eta = -5$(DFS). Inspecting the percentage of generated CoT steps that continue current optimal

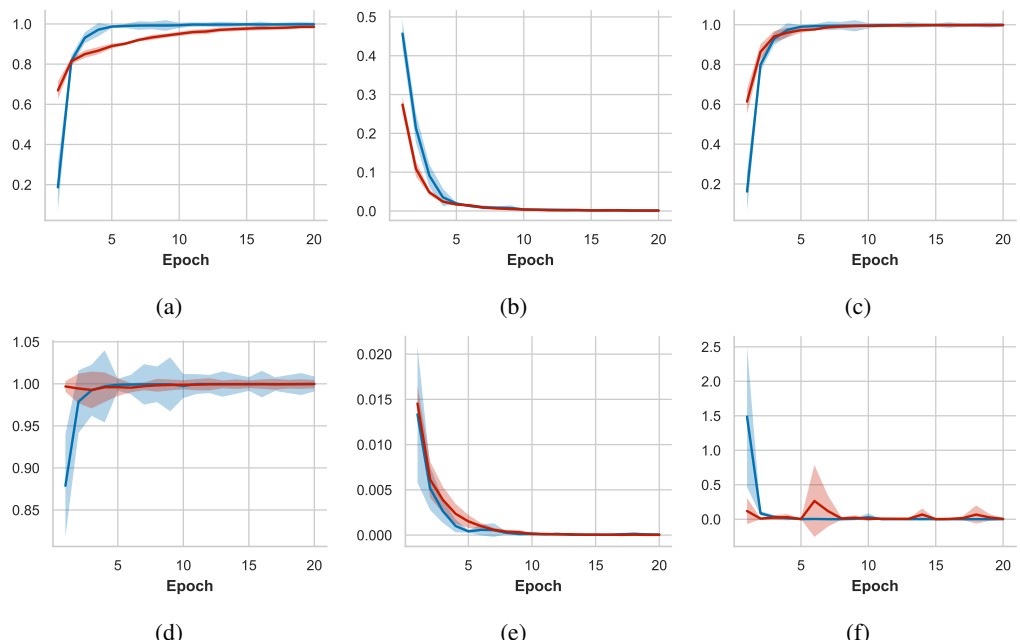

Figure 7: **Reasoning steps metrics.** Comparison of CoT-level metrics between models trained on traces with efficiency $\eta = +5(\text{DP})$ and $\eta = -5(\text{DFS})$. Panels: (a) percentage of subproblem optimal steps, (b) percentage of repeated reasoning steps, (c) percentage of consistent steps, (d) percentage of possible sub-paths, (e) percentage of steps with a skipped subproblem, (f) average numbers of syntax errors.

sub-paths —an optimality constraint that is fulfilled by all training traces—, we see that the inefficient $\eta = -5(\text{DFS})$ model learns more quickly to avoid unnecessary and suboptimal steps, while in the $\eta = +5(\text{DP})$ setting, this metric does not reach full convergence. This indicates that models trained on efficient dynamic programming (DP) traces may still occasionally select suboptimal paths, possibly due to the random order in which same-level paths are explored in $\eta = +5(\text{DP})$, making their relative positions in the trace less predictable.

As an illustration, we show two correct traces below, produced for the same graph. While $\eta = -5(\text{DFS})$ builds trains-of-thought by chaining depth-first-search moves (here we highlight one in blue)

```
BoT n0 n2 2 | n0 n2 n5 4 | n0 n2 n5 n8 6 | n0 n2 n5 n8 n9 8 | n0 n2 n5 n7 5 | n0 n2 n5 n7 n9 6 |

    n0 n2 n4 3 | n0 n2 n4 n7 4 | n0 n2 n4 n7 n9 5 | n0 n2 n4 n8 5 | n0 n2 n4 n8 n9 7 | n0 n3 1 |

    n0 n3 n5 2 | n0 n3 n5 n7 3 | n0 n3 n5 n7 n9 4 | n0 n3 n5 n8 4 | n0 n3 n5 n8 n9 6 | n0 n3 n4 2 |

    n0 n3 n4 n8 4 | n0 n3 n4 n7 3 | n0 n1 2 | n0 n1 n6 3 | n0 n1 n6 n8 4 | n0 n1 n6 n7 4 | EoT ,
```

the succession of steps in the $\eta = +5(\text{DP})$ trace breaks the continuity of the path composition (here highlighted in red), reducing the auto-correlation of the sequence of steps:

```
BoT n0 n2 2 | n0 n1 2 | n0 n3 1 | n0 n1 n6 3 | n0 n3 n5 2 | n0 n2 n4 3 | n0 n3 n4 2 | n0 n2 n5 4 |

    n0 n1 n6 n8 4 | n0 n3 n4 n7 3 | n0 n3 n5 n8 4 | n0 n3 n4 n8 4 | n0 n3 n5 n7 3 | n0 n1 n6 n7 4 |

                    n0 n3 n4 n7 n9 4 | n0 n1 n6 n8 n9 6 | EoT .
```

## B.2   Additional statistics on CoTs

We further analyze the effects of varying the efficiency temperature $\eta$ on the Chain-of-Thought (CoT) length. Fig. 8 shows histograms of the number of CoT steps for $\eta = \pm 5$. Notably, the distribution for $\eta = -5(\text{DFS})$ exhibits a broader range of step counts, suggesting intuitively that the $\eta = +5(\text{DP})$ traces should be easier to fit.

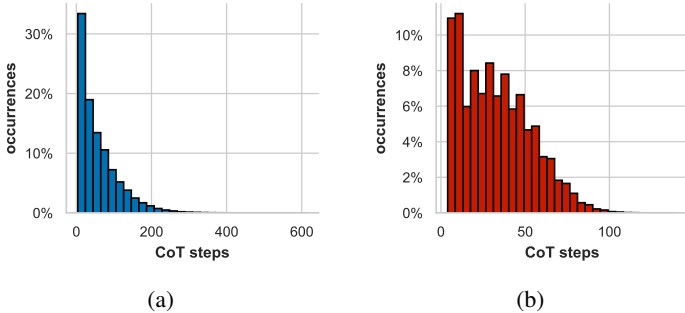

(a)                                                  (b)

Figure 8: Ground truth CoT length in tokens. (a) $\eta = -5$(DFS), (b) $\eta = +5$(DP)

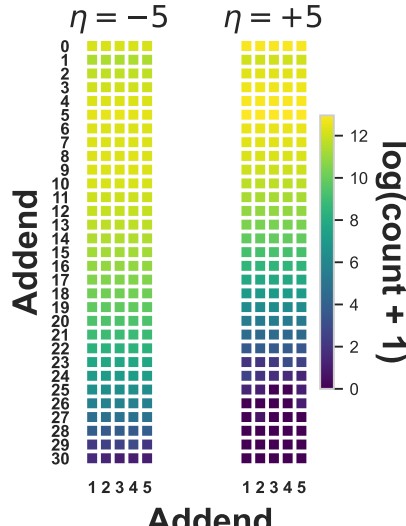

Figure 9: Distribution of integer addition.

Similarly, Fig. 9 illustrates the distribution of integer additions encountered in a typical training set for the two types of traces. While both models receive more than sufficient examples in the lower values of the matrix, where the bulk of the addition operations take place, it is clear that the model trained with $\eta = +5$(DP) requires fewer addition operations to learn compared to $\eta = -5$(DFS), which would point to a simpler task acquisition for the efficient setting and, in principle, give the (DP) approach an advantage in terms of learnability.

These additional statistics further underscore the critical importance of reasoning trace structure for effective learning.

### B.3    Impact of model size and data scale

The experiments presented in the main text considered a transformer architecture with 3 layers, and training datasets containing 32M-128M tokens. In this section, we explore the impact of switching training configurations by varying the number of model layers and dataset sizes. To facilitate training on larger architectures, in this comparison we adopt a standard learning rate scheduler, linearly annealing a starting learning rate of $5 \times 10^{-4}$ down to 0 without warmup.

We report the results in the $\eta = +5$(DP) setting, exhibiting the largest performance variation. As shown in Fig. 10(a), the generalization of the 3-layer model (solid line) reaches a better peak performance compared to the fixed learning rate training, but incurs high variance. As expected, with increased computational power, the larger 6-layer model (dashed line) improves the best performance, reaching levels comparable to $\eta = +5$(DP), 3 layers and 128M tokens, in Fig. 4(a).

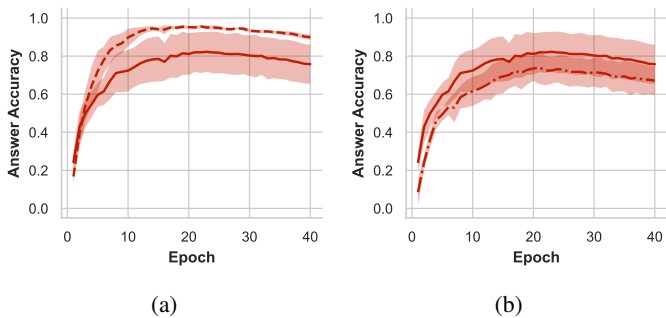

Figure 10: **Ablation studies.** (a) Comparison of the generalization performance between a 3-layer model (full) and a 6-layer model (dashed) trained on $\eta = +5$(DP) with 32M tokens. (b) Comparison between 3-layer models trained on 16M (dash-dot) and a 32M (full).

We note that, while the addition of the scheduler introduces greater variance in the training runs, the relative advantage of the $\eta = -5$(DFS) model's generalization over the $\eta = +5$(DP) model, presented in the main text, is preserved.

Finally, in Fig. 10(b), we show the impact of halving the training budget to 16M, with a sensible 10% decrease of performance on average.

