# OpenReview forum: "On the Bias of Next-Token Predictors Toward Systematically Inefficient Reasoning: A Shortest-Path Case Study"
_NeurIPS.cc/2025/Conference — NeurIPS 2025 poster_

### Official Review · Reviewer_Zh3k · 2025-06-18

**Clarity:** 3
**Significance:** 3
**Originality:** 3
**Rating:** 5
**Confidence:** 4

**Summary:**

This work aims to understand how chain-of-thought reasoning traces and test-time compute aid reasoning performance. More specifically, they aim to understand the impact of the structure, efficiency, and length of reasoning traces on performance of a model trained from scratch on a shortest-path graph task. The authors make several observations: reasoning traces enable models to do this task at all for deeper graphs (directly outputting the shortest path works well only for short graphs). Then, inefficient depth-first search style reasoning traces work better than those reflecting an optimal dynamic programming solution. The authors attribute this to the fact that the reasoning steps follow each other more directly there, with less specific computation that has to be done for each step (each step easier to predict). They further study the impact of test-time compute through deterministic redundancy (if the model can predict where the redundancy occurs, it can use that to do additional computations). This does not help, meaning the actual steps you output matter. Finally, non-systematic structure in the reasoning traces works worst, again attributed to fact that the model cannot predict the next token accurately here (similarly for randomized redundancy). This indicates structure over content is more important in the reasoning trace.

**Questions:**

- The list of types of reasoning traces is a bit hard to follow, after continuing reading the paper I understood but perhaps you could already annotate which of each is the DP and DFS one (l133-l142)
-  Line 210-211; to say that more test-time compute doesn't help, you would have to compare on graphs of the same length, do you do that? I don't see how you can get this conclusion from the $\eta=0$ curves in fig 4.

**Ethical Concerns:**

["NO or VERY MINOR ethics concerns only"]

**Final Justification:**

I remain of the opinion that we should accept this paper because the paper has interesting findings that contribute to our knowledge using a sound experimental setup. Particularly, that inefficient reasoning works better than efficient, tied to the fact that predicting the next token is easier there.

**Limitations:**

Mostly properly addressed, but could discuss how training from scratch might give different results than test-time compute on pretrained models.

**Paper Formatting Concerns:**

N.A.

**Quality:**

3

**Strengths And Weaknesses:**

Strengths:
- Interesting and novel finding that inefficient reasoning works better than optimal reasoning traces, convincingly tied back to the fact that the intermediate steps in the inefficient traces follow more naturally from each other (each token easier to predict)
- The paper is very clearly written, with good figures
- Sound experimental setup,

Weaknesses:
- The results and findings would be stronger and tie more to real-world LLMs if experiments with pretrained models were done. For example through finetuning on the graph task with different style reasoning traces (essentially the same thing as done here but with pretrained models).
- The findings would also be strengthened when an additional task would've been investigated. Right now it's unclear whether the findings would generalise beyond graph-based tasks where things like depth-first search are less obviously applicable (e.g. countdown).

---

> ### Author Rebuttal · Authors · 2025-07-31
>
> We thank the reviewer for their constructive feedback. We are pleased that they found our research novel and interesting, the paper clearly written, and the experimental setup sound. Below we address their concerns in detail.
>
> **Weaknesses**
>
> We agree that exploring the applicability of our findings to real-world LLMs is a compelling direction. We chose not to fine-tune pretrained models for this study due to several challenges: 1) Lack of control over pretraining data: pretrained models may have been exposed to similar algorithmic problems, introducing confounding factors and biases toward particular solutions (DP algorithm in this case, since it is optimal for this problem); 2) such models have their own tokenizer which necessarily differs from ours; 3) the fine-tuning process would be quite expensive on a large model considering that our models necessitate a rather large training corpus (roughly 250K training traces) to learn the task syntax and generalize reasonably.
>
> To overcome at least the latter limitation, few shot learning could be a possible alternative. However, considering the diversity of the graphs in the dataset, various examples per graph depth need to be incorporated in the context. Given that the average length for a $\eta=-5$(DFS) example with graph depth 7 is 1228, and 849 for depth 6, the context length utilized by the few-shots grows fast. This may give rise to various issues as information retrieval and understanding is known to get harder as the size of the context increases.
>
> In our work, we decided to contribute to the research line established by recent works (Lehnert et al. (2024),  Su et al. (2025), Bachmann et al. (2024)) employing algorithmic tasks to train transformers from scratch to elucidate some fundamental properties of these models. Exploring non-DP algorithmic tasks is certainly a valuable direction that we intend to explore in the future. However, extending our framework to novel algorithmic problems is not immediate as it would require the design of brand-new tokenizers and corresponding metrics to assess the quality of the trained models, and a new recipe for controlling the trace efficiency. In the spirit of a proof-of-concept, the choice of our tasks was driven by the necessity to identify a framework that was well-aligned with the research question we intended to probe, i.e. the controlled exploration of the dependency of next-token-predictors on the structure and efficiency of CoT traces.
>
> **Questions**
>
> - We acknowledge that the clarity of the list of reasoning types can be improved. Following the reviewer’s suggestion, we have modified our manuscript to make the link between the $\eta$ parameter and the corresponding algorithmic strategy clearer and more explicit from the beginning of the paper.
> - We apologise to the reviewer for the confusion. As can be observed from Fig 2(b), $\eta=0$ training traces are longer on average than those for $\eta=+5$(DP). Nevertheless, the model trained with $\eta=+5$(DP) outperforms the one trained with $\eta=0$ traces, thus allowing us to conclude that “more test-time compute –here the length of the trace– does not necessarily imply better test performance”. We are going to rephrase it to improve clarity.
>
> **On the limitations**
>
> We acknowledge that training from scratch differs from inference-time manipulation of pretrained models. While we briefly mention this, we will expand the discussion to emphasize the potential implications of this difference.
>
> We thank the reviewer for the time spent reading and reviewing our manuscript. We remain available to further discuss any remaining questions/doubts the reviewer may have.
>
>
> **References**
>
> - Lehnert, Lucas, et al. "Beyond a*: Better planning with transformers via search dynamics bootstrapping." arXiv preprint arXiv:2402.14083 (2024).
> - Su, DiJia, et al. "Dualformer: Controllable fast and slow thinking by learning with randomized reasoning traces." arXiv preprint arXiv:2410.09918 (2024).
> - Bachmann, Gregor, and Vaishnavh Nagarajan. "The pitfalls of next-token prediction." arXiv preprint arXiv:2403.06963 (2024).

---

> > ### Comment · Reviewer_Zh3k · 2025-08-01
> >
> > Thanks for the responses, they address my weaknesses and I agree as it stands "the controlled exploration of the dependency of next-token-predictors on the structure and efficiency of CoT traces" is a good contribution to the field. I will remain my positive score.

---

### Official Review · Reviewer_JctD · 2025-07-03

**Clarity:** 3
**Significance:** 2
**Originality:** 3
**Rating:** 5
**Confidence:** 4

**Summary:**

The authors build a sandbox in which a small decoder-only transformer (Phi-3) must solve shortest-path problems on layered graphs. The transformer is then trained on `question -> trace -> answer` triples and compare the effect of **efficient traces** (in which the traces are optimal bottom-up DP) vs. **inefficient traces** (in which the traces involve backtracking). Finding that the **inefficient traces** lead to better generalization, the authors argue that step-by-step backtracking better matches the inductive bias of autoregressive LMs.

**Questions:**

1. Have the authors tried the same experiment on another algorithmic benchmark, or on natural-language datasets (e.g., math)? Do the same trends hold?
2. Have the authors tried the same experiment on varying model sizes? Is there a point at which inefficient traces stop being optimal?
3. The eureka jump is intriguing. Have the authors inspected attention patterns / neuron activations before and after the jump for more insight?

**Ethical Concerns:**

["NO or VERY MINOR ethics concerns only"]

**Final Justification:**

The authors addressed my concerns and questions; though I do think another round of revision would be valuable, I have raised my score from 4 to 5.

**Limitations:**

Yes

**Paper Formatting Concerns:**

None.

**Quality:**

3

**Strengths And Weaknesses:**

**Strengths**
- The authors set up a clean, controlled experimental sandbox to study changes in trace structure and length.
- By keeping the total token budget fixed, the authors are able to cleanly focus on changes in trace structure.
- The paper is clear, well written, and original.

**Weaknesses**
- The biggest weakness of this paper is that the experiments are only done on one experimental setting (shortest-path in layered graphs), for one model (Phi3) that is **extremely** small. It is not clear if the inductive bias narrative transfers to models with B's of params and more emergent planning abilities. It is also not clear if the lesson about inefficient traces being more helpful would translate to other tasks.

---

> ### Author Rebuttal · Authors · 2025-07-31
>
> We thank the reviewer for their valuable feedback. We are glad they found our experimental testbed clean and controllable and that the paper has been judged well-written and original. In the following, we address the reviewer’s concerns.
>
> **Weaknesses**
>
> We emphasize that our work is a focused case-study, and we acknowledge its limitations in terms of generality (see Limitations, section 6).  The choice of our experimental setting was mainly motivated by our quest for better understanding how relatively simple transformers develop their reasoning process, and which traces such models ultimately privilege. To probe these research questions, we designed an ad-hoc tokenizer and introduced a tunable parameter $\eta$ allowing us to carefully control the level of efficiency and redundancy of the reasoning traces. The proposed setting is *intentionally* kept as simple as possible to focus on the above research questions and factor out eventual confounding effects stemming from complex tasks’ syntax (e.g. math) and eventual biases and leakages inherited from the pretraining of large models.
> Our work fits within the research line established by several recent works studying how transformers behave when tasked to learn and reproduce specific well-known algorithmic routines as proxies of reasoning traces (Lehnert et al. (2024) and Su et al. (2025), Bachmann (2024)). Our results align with these works in terms of the usefulness of learning to produce intermediate reasoning steps as opposed to tasking the model to directly solving the task without any CoT. In addition to that, our specialized setting allows us to investigate how the structure of the CoTs influences performance, an aspect that was not explored in detail in prior works.
> We agree with the reviewer that extending our framework to other (non-DP) tasks represent an interesting research direction. However, such an extension is not immediate as it would require finding similar mechanisms to control the efficiency of the trace and it would necessitate the development of a brand-new tokenizer and synthetic data generator along with specialized metrics to evaluate the models’ quality. For the sake of clarity, we decided to focus on a single task as we believe it nicely encapsulates all the properties we needed to probe.
> Regarding the model size, we selected a minimal working configuration, as is common in many similar studies. Given the training context length, we focused on graphs for which all types of algorithmic traces would fall within the token limit, and then on a model size sufficient to observe good generalization with the Q-CoT-A approach, but not in the Q-A approach. In essence, we chose the smallest model that still allowed the key phenomena to be clearly observed while keeping the experiments lightweight. Nonetheless we trained models as big as 235M parameters using both Phi3 and GPT2 architectures (see answer to Question 2 below for more details) with no substantial dependency on the architectural details in terms of the final results. While no testing has been performed on models with billions of parameters, it can be observed, from reasoning-oriented LLMs such as Deepseek-R1, that even at these large sizes LLMs typically exhibit highly inefficient reasoning traces. This observation was one of the main motivations of our study, i.e. characterizing the relationship between next-token-predictors and the efficiency of the traces they are trained on.
>
> **Questions**
>
> 1. The core motivation of our work is to investigate how the structure and efficiency of reasoning traces influence learning dynamics in next-token prediction. To isolate these factors, we chose a task that offers precise and continuous control over trace efficiency through a single tunable parameter ($\eta$). This level of control is difficult to replicate in natural language or math benchmarks, where answer formats tend to be standardized and optimized for clarity—often featuring a single, highly efficient solution per question. In such domains, generating multiple semantically equivalent but structurally different reasoning traces (with varying efficiency levels) at scale would require significant manual effort or sophisticated paraphrasing/generation pipelines, which may introduce confounds such as semantic drift or unnatural phrasing. Similarly, algorithmic tasks beyond shortest paths (and where DP is not applicable) would require careful task selection and potentially the development of new tokenizers and evaluation metrics to enable comparable experimental control, as well as a new recipe for controlling the trace efficiency. That said, we agree that extending the study to additional algorithmic domains or natural language settings would be a valuable next step. We see this as a promising direction for future work and believe that our methodology could help inspire ways to structure and probe reasoning traces in more complex environments.
> 2. We thank the reviewer for their question. Yes, we ran experiments comparing $\eta=-5$ (less efficient traces) and $\eta=+5$ (more efficient traces) across different model sizes and graph complexities. We observed that as model capacity increases relative to task difficulty (e.g., larger models on simpler graphs), both models eventually achieve 100% test accuracy. However, models trained with $\eta=−5$ consistently converge faster than their $\eta=+5$ counterparts across the board.  Our hypothesis is that the layer-by-layer exploration (of the more efficient traces $\eta=+5$) induces a higher degeneracy in the exploration order and requires a more flexible internal circuit tasked to identify the partial path to be continued, since the relative positions of different path continuations are less predictable.  To your question about whether there's a point where inefficient traces stop being optimal: in our experiments, we did not find a reversal where more efficient traces became better even in terms of convergence speed. However, at higher capacities both trace types eventually perform equally well at convergence. We believe investigating whether a crossover point, where efficiency becomes favourable exists, is an interesting direction for future work.
> 3. We also find the jump very intriguing! We have confirmed this behaviour on more runs per experiment, increasing the number of seeds from 3 to 5. Our current smallest model has 3 layers, 12 heads and ~28M parameters. Our plans for future work is to reduce the size of the model further to facilitate mechanistic interpretability and see if some circuits emerge right after the loss transition point.
>
>
> We sincerely thank the reviewer for the time spent reading our paper. We would be happy to further discuss any remaining concerns and would be grateful if the reviewer would consider adjusting their score should our responses satisfactorily resolve their questions.
>
> **References**
>
> - Lehnert, Lucas, et al. "Beyond a*: Better planning with transformers via search dynamics bootstrapping." arXiv preprint arXiv:2402.14083 (2024).
> - Su, DiJia, et al. "Dualformer: Controllable fast and slow thinking by learning with randomized reasoning traces." arXiv preprint arXiv:2410.09918 (2024).
> - Bachmann, Gregor, and Vaishnavh Nagarajan. "The pitfalls of next-token prediction." arXiv preprint arXiv:2403.06963 (2024).

---

### Official Review · Reviewer_ocpL · 2025-07-03

**Clarity:** 2
**Significance:** 3
**Originality:** 4
**Rating:** 5
**Confidence:** 3

**Summary:**

In a carefully controlled setting, the authors investigate the reasoning behavior of modern LLMs. By systematically altering the reasoning trajectories in training samples, they observe the models’ learning dynamics. They arrive at a surprising conclusion: LLMs are better at following structured, step-by-step reasoning, rather than finding creative shortcuts to solve problems.

**Questions:**

1.How was the graph data used in training and testing constructed? In particular, were there any rules governing how nodes were connected? I’m concerned the model may have learned node-connectivity patterns rather than actual reasoning rules.

2.The study focuses solely on synthetic data. Do the authors expect their findings to hold on publicly available datasets? Have they tested whether the same bias appears there?

**Ethical Concerns:**

["NO or VERY MINOR ethics concerns only"]

**Final Justification:**

The authors’ response resolved my confusion regarding the dataset generation process and also provided valuable insights into the interpretability of the experimental results. Although the work lacks some comparative validation on public datasets, I still consider it a solid piece of work.

**Limitations:**

yes

**Paper Formatting Concerns:**

No format issue.

**Quality:**

3

**Strengths And Weaknesses:**

Strengths:

1.The custom tokenizer designed for the shortest-path problem, along with the experimental setup, is very novel.

2.The study offers valuable insights into how Chain-of-Thought training data can be constructed in specialized scenarios.

Weaknesses:

1.The conclusions are based on a single synthetic dataset, which may be too narrow. The highly structured nature of the input makes the task closer to sequence prediction than genuine reasoning.

2.The explanation for why structured but inefficient reasoning outperforms optimal strategies is not entirely convincing. The weaker performance of dynamic programming (DP) traces may result from limited coverage in training, as DP trajectories are less repetitive and harder to learn than DFS ones. The better results with DFS may simply reflect that such traces are more learnable for LLMs.

---

> ### Author Rebuttal · Authors · 2025-07-31
>
> We thank the reviewer for the provided feedback. We are glad to hear that the originality of our approach has been appreciated and that our framework offers a valuable setup to generate CoT data for controlled experimental analysis. In the following we address the points raised by the reviewer.
>
> **Weaknesses**
>
> 1. Multiple definitions of reasoning exist, as there is no universally accepted technical definition. We believe our task (finding the shortest path in a graph) falls within the scope of reasoning. When a transformer is given a structured description of a layered graph and asked to identify the shortest path, it must: (i) integrate information across layers and develop a high-level strategy to solve the task (ii) simulate a traversal process akin to DFS or dynamic programming, gradually building up the path leading to the final answer and involving the solution of intermediate subtasks (iii) compare and evaluate alternative paths. These requirements would be commonly associated with reasoning tasks in the community.  We also point out that using algorithmic traces to model reasoning in transformers represents a strategy adopted by other recent works in the community (Lehnert et al. (2024) and Su et al. (2025), Bachmann (2024)). Our framework introduces multiple possible algorithmic reasoning strategies to solve the considered task and studies which one of them is more aligned with the inductive bias of transformers, hence providing insights into the mechanisms behind the development of CoTs in such models.
> 2. We agree that different coverage between $\eta=+5$ and $\eta=-5$ could have undermined our findings. However, in section B2 in the supplementary material we show that $\eta=+5$ (DP) has actually *more* coverage than $\eta=-5$ (DFS), as the former setting implies learning fewer additions and the associated sample length distribution is more concentrated compared to $\eta=-5$ (DFS). This should, in principle, give the $\eta=+5$ (DP) approach an advantage in terms of learnability. In addition, when the two approaches are compared at a fixed token budget, $\eta=+5$ (DP) has *more* training samples than $\eta=-5$ (DFS), as the former is generally characterized by shorter CoTs. Nevertheless, the DFS trace setting consistently demonstrates better generalization. We will make sure to better highlight these points in the main body of the revised version of the paper.
>
> **Questions**
>
> 1. Graphs are generated by sampling uniformly the number of layers and nodes in each layer. For every possible edge between two nodes in consecutive layers, the edge is added to the graph with a probability $p$ and its cost is sampled uniformly. We point the reviewer to section A2 of the supplementary material for more details on the graph generation. A lower bound estimate of the number of unique graphs possible in our setting is $5^{180}$. As such, we don’t believe node-connectivity can be robustly used in our setting to solve the task.  In addition, graph generation is independent on the $\eta$ parameter, hence the different generalization performance shown across $\eta$ values are comparable and $\eta=-5$ (i.e. DFS-like traces) emerges as a clear winner.
> 2. We emphasize that our work is a focused case-study and we acknowledge its limitations in terms of generality (see Limitations, section 6). Nevertheless, our work fits into a recent research line aimed at studying reasoning in language models in controlled settings where training data can be programmatically generated and the performance of the model easily assessed. Some of our results are aligned with the findings of other works (e.g., Lehnert et al., 2024; Su et al., 2025), such as the importance of intermediate reasoning steps as opposed to directly predicting the answer, hence providing further evidence to existing research. Besides this, our setting allows us to introduce an additional degree of freedom which is the efficiency (and, indirectly, the length) of the reasoning trace (modulated via the parameter $\eta$). This element of novelty introduced by our setup is not trivial to incorporate in other non-DP settings. Furthermore, we note that extending our analysis to other tasks would require non-trivial changes to the framework (e.g. a novel ad-hoc tokenizer and efficiency knob) hence incurring in the risk of reducing clarity. Finally, existing benchmarks and publicly available datasets do not have the same degree of controllability as our synthetic setup. In addition, they are typically intended for fine-tuning pre-trained LLMs or for use in a few-shot setting. Since such models have been exposed to vast and opaque pre-training corpora, we cannot rule out the possibility that algorithmic strategies such as dynamic programming were already internalized by the model during training. This lack of control introduces confounding factors that our synthetic setup explicitly avoids.
>
> We sincerely thank the reviewer for the time spent reading our paper. We would be happy to further discuss any remaining concerns, and we would appreciate if the reviewer could consider raising their score in case our responses address their questions adequately.
>
>
> **References**
>
> - Lehnert, Lucas, et al. "Beyond a*: Better planning with transformers via search dynamics bootstrapping." arXiv preprint arXiv:2402.14083 (2024).
> - Su, DiJia, et al. "Dualformer: Controllable fast and slow thinking by learning with randomized reasoning traces." arXiv preprint arXiv:2410.09918 (2024).
> - Bachmann, Gregor, and Vaishnavh Nagarajan. "The pitfalls of next-token prediction." arXiv preprint arXiv:2403.06963 (2024).

---

> > ### Comment · Reviewer_ocpL · 2025-08-07
> >
> > Thank you for your explanation of the experimental results. I think this is a very good piece of work. However, it is somewhat lacking in comparisons on public datasets. The authors you cited in [39] and [40] have proposed a public dataset for algorithmic reasoning called CLRS (The CLRS Algorithmic Reasoning Benchmark, Veličković et al., NeurIPS 2022). You can find it at https://arxiv.org/abs/2205.15659.
> > I hope you can continue exploring inference optimization mechanisms in this direction. All in all, I will raise my positive score.

---

### Official Review · Reviewer_RyFF · 2025-07-04

**Clarity:** 2
**Significance:** 3
**Originality:** 3
**Rating:** 3
**Confidence:** 4

**Summary:**

The authors present a synthetic testbed to test how well transformers can learn to find the shortest path in a directed acyclic graph with different search strategies. They experiment with a less efficient depth-first-search-like strategy and a more efficient dynamic programming strategy. They find through extensive experiments involving training Phi3 models from scratch that transformers are able to better generalize to unseen graphs when they are trained on the less efficient search strategy.

**Questions:**

From Figure 2, it seems that the redundancy setting is not exactly comparable to the \eta = -5 setting: The mean CoT length in the redundancy setting is 12% larger than that of \eta = -5, and the variance is 25% smaller.

Line 188: "...good generalization on the larger graph instances can only be achieved when the model is allowed to produce a reasoning trace."
  Since L = 7 (line 183), the model should have seen graphs with a maximum depth of 7, right? If so, then depth-7 graphs would not be "larger graph instances." Or by "larger graph instances" did you mean the graphs with depths within the training distribution that are larger than other graphs from the training distribution? What is the distribution of graph depth of the training examples?

The analysis on the model's confidence of the next step prediction is unclear: For \eta = 0 and \eta = +5, isn't low confidence expected? If there are multiple valid next steps at a given point in the exploration, then the model should assign uniform probability to each of the valid next steps. In this sense, next token confidence would only be a good proxy for learnability _if the task is not too ambiguous._ How often do the exploration traces encounter points where there are more than one valid next steps (for \eta = -5, 0, and +5)?

Line 221: Why would it necessarily be the case that "incremental steps [immediately building] upon previous steps" would result in higher next-token confidence?


The paper does not make many grammatical errors, but the writing can be much improved. Some of sentences are difficult to read and can be made much clearer. I list (non-exhaustively) some of the issues below and I recommend the authors carefully re-read the paper and correct all such errors. The below list also contains some questions about the content:

Figure 1: The edge costs are drawn as nodes which is misleading. I strongly suggest the authors draw the edge costs as labels rather than nodes.

Line 81: Spurious hyphen?

Lines 87-89: (minor) These lines seem to be spaced much farther apart than other lines.

Figure 2: I believe "Effect of...on..." rather than "Dependency on...of..." is closer to the intended meaning.

Line 115: So does a larger or smaller value of \eta produce more "structurally efficient" reasoning traces? It is not easy to understand the meaning "structural efficiency" if no precise definition is provided aside from the algorithm in Figure 2.

Line 127: How are reasoning steps "disregarded"? Are they randomly removed from the reaoning trace? If so, they would have no effect on the subsequent generation of the transformer, won't they? For systematic redundancy: Is it only possible to increase the length of the CoT by an integer >= 2 (since *each* reasoning step is repeated)?

Line 146: Is this the percent increase or decrease?

Line 171: Duplicate "iv)".

Figure 3a: This plot needs a legend. The colors are also somewhat random. I recommend using a gradient of three colors to better depict the sequence of 3, 5, and 7-depth graphs.

Line 204: Does "dashed" and "full" refer to the plot in Figure 4a? The figure hasn't even been referenced yet, so you should avoid referring the line style in the plot here.

Line 208: "less" -> "fewer"

Figure 4: I strongly suggest that legends be added to the plots in this figure.

Line 210: (minor) Use "---" or \textemdash for em dash, and remove spaces around each dash.

Lines 215-218: This sentence is unclear and difficult to understand. What does "understand" mean here, in the context of "harder to understand and mimic for the trained model"? What does it mean for the model to understand an exploration order?

Line 229: "the model transitions from repeating the same reasoning steps from 20% to 3% of the times" This is ambiguous. Does this mean that, prior to this transition point, the model is repeating steps in 20% of the validation examples? Training examples? Or are 20% of the reasoning steps repeated across all reasoning steps in all validation examples? (or training examples?)

**Ethical Concerns:**

["NO or VERY MINOR ethics concerns only"]

**Final Justification:**

Overall, I do believe the paper could benefit greatly from another round of revision to improve the writing and to incorporate the proposed clarifications in the rebuttal.

One issue the authors resolved during the discussion is why they did not consider testing larger graph depths, as it is not clear whether/how their findings would generalize to larger graphs. They explain that this was due to the context limit of their model, which is 4096 tokens. They were not able to train larger models due to compute limitations. I found this explanation to be satisfactory.

However, one issue that was not addressed was in the experiments comparing the model's accuracy with their next-token confidence. There is a potential confounder here in that the two tasks (DFS and DP) are fundamentally different: In DFS, the "correct" next token is often unique, whereas in DP, there are oftentimes many valid next steps. Thus, due to the difference in tasks, the model trained on DP would necessarily have lower next-token confidence, regardless of which model is more accurate. As a result, I chose not to increase my rating.

**Limitations:**

See Weaknesses above. The authors acknowledge that their testbed is rather artificial and their findings may not necessarily generalize to real-world settings. However, I do believe it represents an interesting exploration in fundamental capabilities of transformers when trained with idealized data (therefore providing insight into the upper limits of transformer capabilities on this task).

**Quality:**

2

**Strengths And Weaknesses:**

Strengths:
 - The testbed reveals interesting behavior where transformers learn to find the shortest task more generalizably using a less efficient search technique.
 - The paper represents an interesting exploration in fundamental capabilities of transformers when trained with idealized data (therefore providing insight into the upper limits of transformer capabilities on this task).

Weaknesses:
 - The exploration is rather limited in terms of graph depth and it is unclear if the findings generalize to larger graphs/CoTs. It would be very interesting to explore the scaling trend of the inefficient vs efficient searching approaches on larger graphs.
 - Some claims are not supported by the experimental results. (see below)

---

> ### Author Rebuttal · Authors · 2025-07-31
>
> We thank the reviewer for taking the time and effort to carefully evaluate our work. We are glad they found our main result interesting and our analysis helpful to study fundamental properties of transformers. In the following, we discuss the raised points.
>
> ---
>
> **Weaknesses**
>
> - Across our experiments we use graphs with depth sampled uniformly from 2 to 7 (included). Together with the other graph generation parameters, the total number of unique graphs that can be sampled is above $5^{180}$.
>   We restricted our analysis to a dataset and model size where the difference between $\eta = +5$ (DP) and $\eta = -5$ (DFS) was apparent. We experimented with other model sizes and dataset difficulty levels. In the biggest tested setting, model-size 235M and graph depth ranging from 2 to 9 (included), we observe that models converge to a generalization accuracy of 100%, nevertheless we still observe the superiority of $\eta = -5$ (DFS) in terms of faster convergence. This trend can also be observed from the comparison between $\eta = +5$ and $\eta = -5$ when both are trained with 128M tokens (Fig 4(a)).
>
> - The doubts on the validity of some claims (W2) are addressed together with the questions.
>
> ---
>
> **Questions**
>
> > From Figure 2, it seems that the redundancy setting is not exactly comparable to the $\eta = -5$ setting...
>
> In the paper, we mainly compare $\eta = +5$ with $\eta = +5$ with redundancy at equal number of tokens in the dataset. The goal of this analysis is to check whether more raw test-time compute (here artificially injected via repetitions) is beneficial. We observe that $\eta = +5$ with redundancy is inferior to $\eta = +5$ without redundancy. This suggests that more test-time compute only cannot explain the performance gain of $\eta = -5$. Note that the runs of $\eta = +5$ with redundancy have both more samples and on average longer CoTs than the $\eta = -5$ runs shown in Figure 1. Nevertheless $\eta = -5$ is still superior.
>
> ---
>
> > Line 188: "...good generalization on the larger graph instances..."
>
> We apologize to the reviewer for the unclear phrasing. In line 188 by “larger graph instances” we refer to graphs that are still within the training distribution. The graph depth in the dataset is sampled uniformly from 2 to 7 included. For further details on the graph generation strategy, we refer the reviewer to section A2 in the supplementary material.
>
> ---
>
> > The analysis on the model's confidence of the next step prediction is unclear...
>
> We agree with the reviewer that our confidence metric has a clear interpretation at a path level, and this was intended. In all of the trace types multiple exploration orders are equally likely and indeed expected that higher degeneracy would ultimately result in reduced predictability in the CoTs. To support this, we report the Shannon’s surprise of the path continuation choice, averaged over the $\eta = -5,+5,0$ datasets:
>
> $$
> \eta = +5 \rightarrow 1.3262 \pm 0.0006; \\
> \eta = -5 \rightarrow 0.4821 \pm 0.0002; \\
> \eta = 0 \rightarrow 1.905 \pm 0.003
> $$
>
> However, the next-token confidence metric was not introduced ad-hoc by our study but is borrowed from Xuezhi Wang et al. (2024) where it is used as a heuristic to establish the quality of an LLM answer. They observe that by sampling multiple answers to a given question, picking the one with higher next-token confidence improved the accuracy compared to randomly picking an answer from the sampled ones. Our goal is to interpret the findings of Xuezhi Wang et al. (2024) in what we believe to be a better controlled environment for their metric. Furthermore, even if the considered metric has a clear interpretation at a path-level, it is not immediately obvious how it translates to the token-level as it can be affected by the next-token prediction steps inside a path.
>
> ---
>
> > Line 221: Why would it necessarily be the case that "incremental steps..."
>
> In the case of $\eta = -5$ (DFS), incremental steps have a high next-token confidence as the exploration order prioritises path continuations that directly extend the last seen partial path.
>
> ---
>
> **Clarity and Stylistic Enhancements**
>
> We greatly appreciate the reviewer’s careful attention to our work and we are going to implement all their recommendations. More specifically:
>
> ---
>
> > I believe "Effect of...on..." rather than "Dependency on...of..." is closer to the intended meaning.
>
> We are going to split the caption into 2 parts, one for (a) and the other for (b). In (a) there is a dependency on $\eta$, as shown in the algorithm, while in (b), in line with the reviewer’s suggestion, there is an *effect* of $\eta$.
>
> ---
>
> > It is not easy to understand the meaning "structural efficiency" if no precise definition is provided...
>
> We agree with the reviewer that the paragraph is not clear and an explicit definition of it is needed. We plan to add the following to the main body:
> “We define structural efficiency as the efficiency, in terms of cot length, given by varying the structure (exploration order) of the algorithm”
>
> ---
>
> > How are reasoning steps "disregarded"?
>
> We apologize for the confusion. By “disregarding with some probability” we mean that, with probability $\geq 0$, we do not remove the sampled step from the queue (while it is added to the trace). As a result, the step can be sampled again in future iterations (possibly multiple times). We will revise the phrasing to improve clarity. The reviewer is right; with the systematic redundancy the length of the CoT can be increased by an integer factor.
>
> ---
>
> > Figure 3a: This plot needs a legend.
>
> We are going to add legends to the plots in Fig. 3 and Fig. 4, improve the colors and fix the “full” and “dashed” reference before referring to Fig. 4(a).
>
> ---
>
> We agree that the sentence:
>
> > “While also the $\eta = 0$ traces are longer than the $\eta = +5$ ones, the associated flat distribution over the exploration order, mixing depth-first and layer-by-layer exploration, is harder to understand and mimic for the trained model.”
>
> is unclear. We intended to say that in the $\eta = 0$ setting the associated flat distribution over the exploration order, that mixes depth-first and layer-by-layer exploration, reduces the confidence of the model in predicting the next step, undermining the learning effectiveness. We will revise it accordingly.
>
> ---
>
> > Lines 215-218: This sentence is unclear and difficult to understand...
>
> We agree that the sentence: “The model transitions from repeating the same reasoning steps from 20% to 3% of the times” is ambiguous. We intended to refer to the average proportion of repeated steps per reasoning trace across all validation samples. We will rephrase it to improve clarity.
>
> ---
>
> We sincerely thank the reviewer for the valuable feedback, which has significantly contributed to enhancing the clarity of our paper. We hope our responses adequately address the raised concerns, particularly regarding the claims perceived as lacking sufficient evidence (W2). We would be grateful if the reviewer could consider revising their score considering these clarifications.
>
> **References**
>
> - Wang, Xuezhi, and Denny Zhou. "Chain-of-thought reasoning without prompting." Advances in Neural Information Processing Systems 37 (2024): 66383-66409.

---

> > ### Comment · Reviewer_RyFF · 2025-08-05
> >
> > I thank the authors for their thoughtful response.
> >
> > > We restricted our analysis to a dataset and model size where the difference between $\eta = +5$ (DP) and $\eta = -5$ (DFS) was apparent. We experimented with other model sizes and dataset difficulty levels. In the biggest tested setting, model-size 235M and graph depth ranging from 2 to 9 (included), we observe that models converge to a generalization accuracy of 100%, nevertheless we still observe the superiority of $\eta = -5$ (DFS) in terms of faster convergence. This trend can also be observed from the comparison between $\eta = +5$ and $\eta = -5$ when both are trained with 128M tokens (Fig 4(a)).
> >
> > Is 9 the largest graph depth in your experiments? Or did you additionally experiment with other dataset difficulty levels? If so, which ones? Were depths larger than 7 not included because the difference between $\eta = +5$ and $\eta = -5$ less pronounced? I believe this would be an interesting result, as it would suggest that the qualitative behavior that you observe at graph depths up to 7 does not necessarily persist at larger graph depths.
> >
> > > We agree with the reviewer that our confidence metric has a clear interpretation at a path level, and this was intended...
> >
> > This motivation for the use of the next token confidence is missing. I urge the authors to strengthen the motivation/rationale for using this metric. In addition, is it not the case that the confidence measure is confounded by the more uniform distribution of next steps in the DP case, as compared to the more "spiky" distribution of next steps in the DFS case?
> >
> > > In the case of $\eta = -5$ (DFS), incremental steps have a high next-token confidence as the exploration order prioritises path continuations that directly extend the last seen partial path.
> >
> > I see. When I was reading the original sentence, I was not considering that it was specifically referring to the DFS case. Without that context, the original sentence is very abstract and difficult to interpret. For example, I am unclear as to the meaning of a "more consequential reasoning process".
> >
> > Overall, I do believe the paper could benefit greatly from another round of revision to improve the writing and to incorporate the proposed clarifications in the rebuttal.

---

> > > ### Author Response · Authors · 2025-08-05
> > > **Reply**
> > >
> > > We thank the reviewer for engaging in the discussion and for their constructive criticism.
> > >
> > > ---
> > >
> > > > Is 9 the largest graph depth in your experiments? ...
> > >
> > > We apologize for the confusion caused by our rebuttal.
> > > What fundamentally limits the size of the graph instances included in our experiments is the context length: even for the largest instances in the training set, we require the \$\eta=-5\$ traces to fit within this constraint to avoid systematic errors for the less token-efficient traces.
> > > In our preliminary experiments, we did consider deeper graphs (up to 9 layers), but also with narrower layers (up to 4 nodes). These graphs were not harder to solve than those explored more thoroughly in the paper (7 layers and 6 nodes per layer), since the \$\eta=-5\$ CoTs were actually shorter on average.
> > > Moreover, two key factors contributed to the closing of the asymptotic performance gap: (1) the increased training data (256M tokens, double the amount used in the paper), and (2) the increased number of transformer layers (we initially used a more standard architecture with 12 layers and 12 attention heads).
> > > Even in our setting, when sufficient data and a larger model are available (see Appendix B.3), the inductive bias can be overcome: the model is, in principle, able to learn any of the considered traces, including DP-like ones.
> > > We emphasize that our study focuses on the interplay between inductive bias and data efficiency, explored through variations in the structure of the reasoning trace. We chose to highlight a regime (minimal data and architecture) in which the different data efficiencies produce a substantial accuracy gap between \$\eta=-5\$ and \$\eta=+5\$. This regime can be sought regardless of the specific graph instances used for training/testing.
> > >
> > >
> > > > This motivation for the use of the next token confidence is missing...
> > >
> > > We thank the reviewer for pointing out this missing motivation.
> > > We will expand this discussion and clarify the choice of metric in the revised version of the paper.
> > > As we attempted to explain in the rebuttal, we anticipated that the higher degeneracy of DP-like traces would correspond to lower model confidence compared to the spikier DFS-like traces—a hypothesis that was confirmed experimentally. This is precisely why our controlled setting, where such a confidence gap is expected and higher next-token predictability was associated with better model performance, can help clarify the heuristic introduced in Xuezhi Wang et al. (2024).
> > > In that study, the metric was motivated by intuitive explanations, but in natural language tasks there was no explicit control over how different answer structures could lead to more forced reasoning paths (i.e., less degeneracy). Therefore, the connection we observe between degeneracy and model confidence is an intended effect—not a confounder.
> > >
> > >
> > > > I see. When I was reading the original sentence,...
> > >
> > > We agree that the imprecise original wording was potentially confusing.
> > > Given the additional space in the final version, we will revise and expand this part, explicitly stating what we mean by the “consequentiality” of the reasoning process and elaborating on the role of degeneracy in the exploration order.
> > >
> > > ---
> > >
> > > We remain happy to further improve our work should additional concerns arise.
> > > We hope that by thoroughly addressing each point and revising the manuscript accordingly, we are not only enhancing the clarity of our work but also reinforcing the soundness of our experimental setup and the robustness of our results and claims. If the reviewer finds these clarifications satisfactory, we kindly hope they may consider adjusting their evaluation accordingly.

---

> > > > ### Comment · Reviewer_RyFF · 2025-08-06
> > > >
> > > > I thank the authors for their response.
> > > >
> > > > > What fundamentally limits the size of the graph instances included in our experiments is the context length: even for the largest instances in the training set, we require the $\eta=-5$ traces to fit within this constraint to avoid systematic errors for the less token-efficient traces. In our preliminary experiments, we did consider deeper graphs (up to 9 layers), but also with narrower layers (up to 4 nodes).
> > > >
> > > > Ah, so how many tokens are generated for inputs of different depths? I assume the context limit is the same as that of Phi3 (128k tokens)?
> > > >
> > > > > We anticipated that the higher degeneracy of DP-like traces would correspond to lower model confidence compared to the spikier DFS-like traces—a hypothesis that was confirmed experimentally. This is precisely why our controlled setting, where such a confidence gap is expected and higher next-token predictability was associated with better model performance, can help clarify the heuristic introduced in Xuezhi Wang et al. (2024).
> > > >
> > > > This still seems like a potential confounding factor: Imagine a task where the next correct token is always unambiguous (e.g., a single path between the start and goal vertices). You could have a model trained to predict the correct token with very high confidence, as well as another model trained to predict the correct token with lower confidence (but it always assigns highest probability to the correct token). But if you use greedy decoding (as you have), both models would achieve the same accuracy on the task, despite the first model having much higher confidence than the second. Thus, model accuracy, next-token confidence, and the number of correct next steps (as you measured via Shannon surprise in your response) are not inherently independent variables.

---

> ### Author Response · Authors · 2025-08-06
> **Comment**
>
> We thank the reviewer for their continued engagement in this process.
>
> > Ah, so how many tokens are generated for ...
>
> For the largest graphs (7 layers and 6 nodes in each layer, generated with a probability ~10^(-5)), the $\eta=-5$ traces have an average length of ~2.7K tokens, with large fluctuations for different exploration orders. In our experiments, the maximum training context length we could employ with our computational resources was 4096. Notice that, in this regime, to include hundreds of thousands of CoT examples, we are already training over several million tokens.
>
> > This still seems like a potential confounding factor ...
>
> We never argued that next-token confidence, the number of correct next steps, and the accuracy are independent variables. Quite the opposite, we are showing that there is a clear correlation between them in our experiments. We agree with the reviewer that, apart from the different degrees of degeneracy in the exploration order, other factors could influence the model confidence, and some of them might be uncontrollable. However, we tried to control for the particular example of spurious effect proposed in the reply above: we are employing the same stopping criterion in all runs, i.e., selecting the checkpoint that achieves the maximum validation accuracy. In the mentioned single-path task, the lower model confidence for the second trained model would need to be caused by a different stopping criterion (as the reviewer said, training "to predict the correct token with lower confidence"), since nothing would prevent the model from becoming as confident as the former one without overfitting. While we cannot claim a causal link between next-token predictability and training effectiveness, we believe their relationship provides a reasonable and intuitive explanation of the observed inductive bias.

---

### Decision · Program_Chairs · 2025-09-17

**Decision:**

Accept (poster)

**Comment:**

The paper studies the efficiency of learning algorithmic reasoning tasks with auto-regressive transformers, using different types of reasoning traces. By considering a controlled setting involving shortest-path problems, the authors find that training with "inefficient" reasoning traces (i.e. resembling DFS-style strategies) can lead to better learning compared to "efficient" traces (e.g. using DP-style strategies, which lead to shorter traces), in the sense that the model learns more quickly and generalizes better to larger graphs. While the setting is somewhat narrow, and some of the claims could be strengthened in the final revision (in particular, see the comment on confidence metrics for DFS vs DP by reviewer RyFF), overall, the reviewers agreed that this is an interesting and important question, and that the experiments thoroughly validate this hypothesis. I thereby recommend acceptance.